# Learning Explicit Semantic-Spatial Synergy for Weakly Supervised Referring Image Segmentation

## Abstract

Weakly Supervised Referring Image Segmentation (WSRIS) aims to segment target objects specified by natural language expressions using only image-text pairs. While recent advances have improved semantic grounding, existing methods still lack explicit mechanisms to incorporate spatial understanding, preventing them from achieving semantic–spatial synergy and leaving a large gap to fully supervised approaches. To address this limitation, we propose $ES^3Net$, a novel framework that **E**xplicitly learns **S**emantic-**S**patial **S**ynergy from the following three perspectives: **First**, we propose an Explicit Spatial Enhancement Module that integrates mask-grounded semantic features with object-centric 3D coordinates derived from readily obtained depth. This produces embeddings where spatial geometry is semantically anchored, enabling accurate localization of position-sensitive expressions. **Second**, a Language Consistency Module is proposed to enforce consistent alignment across diverse expressions referring to the same instance, improving robustness to linguistic variations. **Finally**, we introduce a Confidence-Aware Dense Distillation strategy that transforms high-confidence grounding predictions into pseudo labels, allowing a lightweight student–teacher RIS model to be trained for stable learning and efficient inference. Extensive experiments on RefCOCO, RefCOCO+, and RefCOCOg demonstrate that $ES^3Net$ establishes new state-of-the-art performance, underscoring the importance of explicit semantic–spatial synergy in advancing WSRIS. Source code and models will be released upon acceptance.

## 1 Introduction

Referring Image Segmentation (RIS) is the task of segmenting objects or regions in an image specified by a textual description, requiring two core capability for the model: (**i**) Interpreting the referring text to comprehend thus locate the correct instance/region(s); (**ii**) Segmenting the target region/instance(s). Most existing RIS methods require massive human-annotated pixel-level mask labels as supervision, which has several drawbacks: (**i**) these dense labels can be costly to obtain in the real world. (**ii**) Fully supervised models are likely to overfit human annotation styles. Recently, there has been a growing interest in developing weakly supervised approaches for RIS that rely on readily available image-text pairs, which eliminates the need for precise pixel-level labels. This paper focuses on weakly supervised RIS (WSRIS) using only image-text pairs.

The key challenge of WSRIS lies in accurately and subtly aligning linguistic cues with pixel-level visual features to identify and separate the specific target object from potential similar objects and background without ground-truth label supervision. Inspired by weakly supervised semantic segmentation (WSSS), common practices for previous WSRIS works are to adopt contrastive learning to establish patch-level image-text correspondences. Earlier methods, such as SaG (Kim et al., 2023) and TRIS (Liu et al., 2023), utilize attention-based methods to obtain a heat map and thus extract masks. These works demonstrate a certain capability to understand the referring text. Since SAM has emerged as a universal segmentor to extract precise masks, PPT (Dai & Yang, 2024) and PCNet (Yang et al., 2024b) leverage SAM as a refinement engine, usually extracting the high-response region on the heatmap and using it as a prompt to generate the final fine-grained prediction.

Despite advances, current SoTA methods in WSRIS still significantly fall short compared to their fully supervised counterparts, often due to the absence of explicit semantic–spatial synergy. We analyze this gap from the following perspectives: (**i**) Some approaches (Kim et al., 2023; Liu et al., 2023) rely on the model's implicit attention mechanism alone to associate text with vision, and often fail to perfectly align linguistic hints with the target instance and obtain its segmentation with a precise boundary. (**ii**) Current methods might struggle to interpret geometric cues in referring expressions and to model relationships among multiple instances within the image. In WSRIS, where no explicit

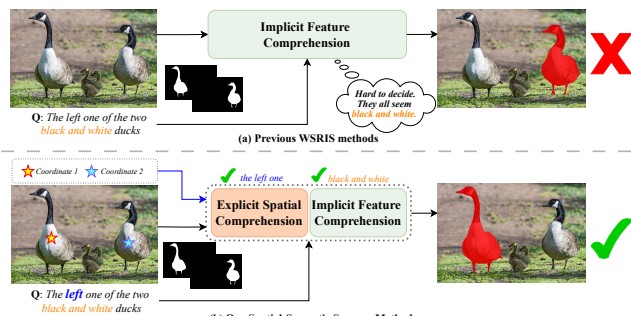

Figure 1: Illustration of (a) conventional Weakly Supervised Referring Image Segmentation (WSRIS) methods, which rely solely on implicit feature learning and often fail to interpret spatial language (*e.g., "left"*), and (b) our spatial-semantic synergy strategy. We enhance the model with explicit spatial awareness, enabling explicit spatial understanding over geometric and directional cues for accurate referring image segmentation.

pixel-level supervision is available, learning to distinguish spatially symmetric phrases such as *"man on the left"* versus *"man on the right"* solely through contrastive learning is particularly challenging due to the inherent symmetry of such concepts in the semantic embedding space. (**iii**) Many frameworks rely heavily on the implicit knowledge encoded in pre-trained vision language models (VLMs), which are trained with image-text pairs and thus lack fine-grained capacity to comprehend complex spatial or semantic relations among multiple instances (*i.e, "the man on the left of the woman"*). Although some methods attempt to address this limitation by introducing auxiliary modules (*e.g.*, spatial rectifiers), such designs often depend excessively on hand-crafted priors, compromising their generalization ability and scalability.

These observations point to a fundamental limitation: current WSRIS frameworks lack an explicit mechanism to achieve semantic–spatial synergy. We argue that beyond semantic alignment, explicit spatial encoding is essential for disentangling geometric relations and for enabling robust grounding under weak supervision. To explicitly address the limitations in spatial understanding in WSRIS, we propose a lightweight but effective **Explicit Spatial Enhancement Module (ESEM)** that explicitly inject the geometric context into the embedding space. By providing explicit spatial encoding, ESEM enables the model to distinguish subtle geometric variations in referring expressions, such as *"left right"* or *"front behind"*, which cannot be reliably captured by implicit attention alone. Our experiments show that this explicit design yields substantial improvements, boosting the model's performance across mainstream datasets. To further facilitate the model's robustness against linguistic variation, we propose a **Language Consistency Module (LCM)** that enforces consistency across diverse expressions referring to the same instance. This design strengthens semantic–spatial synergy by ensuring the model reacts coherently to different yet semantically equivalent descriptions, thus mitigating the instability caused by ambiguous phrases. Finally, we extend the alignment stage into a distillation process that transforms the grounding results into reliable supervision, and propose a **Confidence-Aware Dense Distillation (CADD)** strategy. More specifically, we select mask predictions weighted by their confidences from the alignment model as pseudo labels, which are then used to train a lightweight RIS framework following the student-teacher learning strategy. In this way, knowledge is distilled from the alignment model into a compact segmentor, enabling efficient inference while preserving semantic–spatial synergy. More importantly, this refinement bypasses the limitations of relying solely on SAM-generated masks, where the referred instance may be absent, by grounding supervision directly in the model's own predictions.

The main contributions of this paper are summarized as follows: (**i**) We present a novel WSRIS framework, **ES³Net**, that explicitly learns semantic-spatial synergy, progressively bridging grounding perception and dense segmentation. (**ii**) We propose an Explicit Spatial Enhancement Module (ESEM), which incorporates explicit spatial encoding to complement semantic features. This provides an effective mechanism to strengthen spatial understanding in WSRIS, enabling the model to better deal with location-sensitive expressions. (**iii**) We introduce a refinement stage (CADD) that distills high-confidence grounding predictions into a lightweight RIS model, enabling

efficient inference while avoiding the limitations of SAM-based refinement. (**iv**) We introduce a Language Consistency Module (LCM) that enforces consistent alignment across diverse expressions referring to the same instance, further enhancing the model's robustness to linguistic variations. (**v**) Extensive experiments demonstrate that the proposed method achieves state-of-the-art WSRIS results on RefCOCO, RefCOCO+ and RefCOCOg. The results remain competitive even compared to the oracle fully-supervised model. We also present a first attempt to formalize WSRIS under a more fine-grained criterion: Single-Text Supervision and Multi-Text Supervision, revealing new insights into dataset-level supervision.

## 2 RELATED WORK

**Referring Image Segmentation (RIS).** Referring Image Segmentation (RIS) aims to segment a specific region given a referring text as input. Hu et al. (2016) proposed the first RIS method with CNN-LSTM based encoders. Many methods (Liu et al., 2017; Margffoy-Tuay et al., 2018; Ye et al., 2019) have been conducted since then, mainly focusing on cross-modal visual-language fusion strategies. In recent years, more and more works (Wang et al., 2022; Yang et al., 2022; Dai et al., 2025; Yu et al., 2025) have shown great interest in exploring the cross-modal attention mechanism and transformer architecture to model the long-range dependencies to achieve hierarchical pixel-level alignment. For example, LAVT (Yang et al., 2022) employs language-aware vision transformer architectures to encode visual features with their linguistic context at any spatial location in each transformer layer. Other methods (Yu et al., 2025; Dai et al., 2025) apply BEiT-3 (Wang et al., 2023) based shared vision-language encoder in RIS for deeper integration of visual and linguistic features. Despite advances, these methods require solely on human-annotated segmentation labels for supervision, which is at high cost.

**Weakly Supervised Referring Image Segmentation (WSRIS).** Recently, weakly supervised learning (Xu et al., 2022b) for RIS has attracted more and more attention since it significantly alleviates labor for human annotation. Feng et al. (2024) proposes the first WSRIS method using bounding box annotation supervision. Subsequent works (Lee et al., 2023; Liu et al., 2023; Kim et al., 2023) explore leveraging weaker supervision, where only image-text pairs are introduced. Lee et al. (2023) first extracted the salient image regions corresponding to each word in the sentence by applying the explainability tool Grad-CAM (Selvaraju et al., 2017) on a VLM, then proposed to enhance the semantic relationships between these saliency maps by enforcing intra-chunk and inter-chunk consistency. Liu et al. (2023) instantiates a direct text-to-patch alignment by classifying and contrasting the target-related text and target-unrelated text. Kim et al. (2023) associated texts with image regions through top-down and bottom-up attention mechanisms. However, relying on implicit attention often fails to properly align the linguistic hints with the target instance. Recent WSRIS approaches have also integrated the Segment Anything Model (SAM) (Kirillov et al., 2023) into weakly supervised pipelines, achieving improved results. For instance, PPT (Dai & Yang, 2024) uses CLIP to generate semantic point prompts for SAM and refines them via a multi-step curriculum learning strategy with multi-source pseudo-labels. PCNet (Yang et al., 2024b) leverages SAM-generated masks to impose constraints during progressive text comprehension and visual localization, iteratively refining the response map for point prompting.

Despite these advances, existing methods are limited in that they primarily rely on semantic features from CLIP or SAM, without explicitly encoding spatial features. As a result, they struggle to resolve ambiguities in expressions involving positions. In contrast, our approach introduces explicit spatial embeddings with mask-guided 3D geometry cues, enabling a more effective semantic-spatial synergy that grounds language not only in appearance, but also in spatial position.

## 3 METHODOLOGY

**Overview:** We present a novel Explicit Semantic-Spatial Synergy network (**ES³Net**) for WSRIS. The proposed framework consists of three key components: the Explicit Spatial Enhancement Module (ESEM), the Language-Consistency Module (LCM), and the Confidence-Aware Dense Distillation (CADD) strategy. As illustrated in Figure 2, given an input image-text pair, we additionally generate a set of mask proposals using SAM and the depth image using . We employ two visual encoders: CLIP is applied to the masked image to extract class-level embeddings, while DINOv2

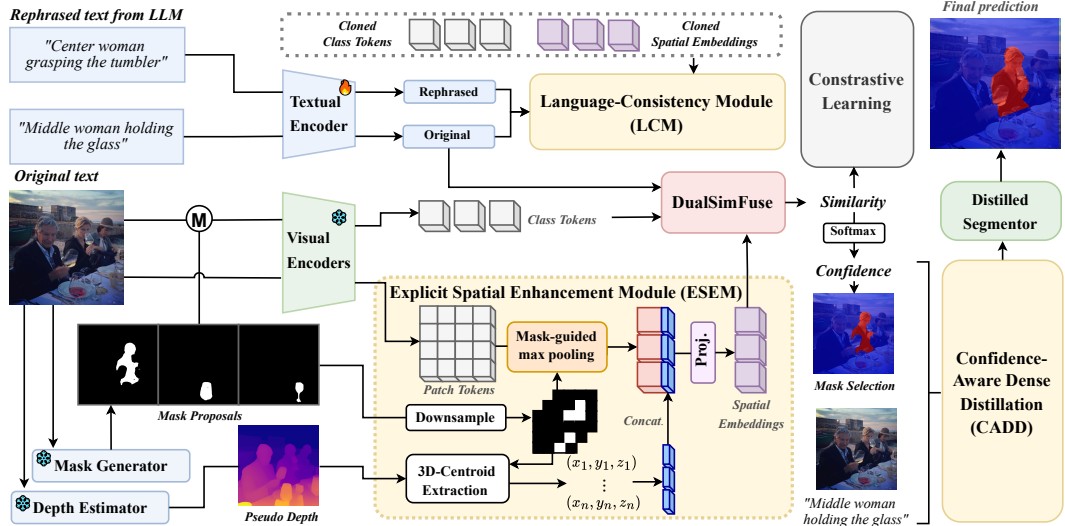

Figure 2: The overview of the proposed **ES³Net**. The Explicit Spatial Enhancement Module is proposed to enhance the semantic features with explicit spatial information. The proposed Language Consistency Module enforces alignment consistency across diverse expressions. The Confidence-Aware Dense Distillation module is introduced to refine dense predictions into a lightweight segmentor for efficient inference. Ⓜ denotes the masking operation.

provides mask-guided patch embeddings that capture detailed appearance cues. We use CLIP text encoder to extract text embeddings. ESEM takes the masked-guided patch embeddings and augments them with explicit spatial cues derived from mask proposals and pseudo depth, generating spatially enhanced embeddings with 3D-aware geometric information. The spatial embeddings, together with the CLIP-derived class embeddings, are jointly aligned with textual embedding in the DualSimFuse module through contrastive learning. The LCM module leverages LLM-rephrased expressions to compute alignment scores with both types of visual embeddings, and enforces consistency by aligning these scores with those from the original texts, thereby improving semantic robustness and reducing ambiguity in language grounding. Finally, we introduce a CADD strategy, which uses the pseudo masks transformed from the alignment results with confidence-based filtering to supervise a lightweight referring segmentation module. This module distills knowledge from a teacher model (*e.g.*, LAVT). In addition to providing efficient inference without extra overhead, CADD also overcomes the limitation of relying on SAM proposals, where missing target masks prevent correct predictions, by enabling the model to generate accurate masks beyond SAM's scope.

## 3.1 DATA PREPARATION AND FEATURE EXTRACTION

**Data Preparation.** Following the conventional practice (Dai & Yang, 2024; Yang et al., 2024b) in the previous work, we use the Segment Anything Model (SAM) (Lin et al., 2024) to generate class-agnostic masks. More specifically, given an RGB image $\mathbf{I}$, we use SAM to generate $N_s$ initial binary mask proposals. For each mask instance, we measure its similarity with the referring text $\mathbf{T}$ based on CLIP. The highly ranked masks are selected for subsequent use. To ensure training efficiency, only the top-N masks are selected, but the model allows any number of masks for inference. Formally, the selection process is formulated as follows:

$$\mathcal{M}' = \underset{\mathbf{M} \in \mathcal{M}}{\text{top-}N} \left( \text{sim} \left( \phi_I(\mathbf{I} \odot \mathbf{M}), \phi_T(\mathbf{T}) \right) \right), \mathbf{M} \in \mathcal{M}' \tag{1}$$

where $\odot$ denotes the Hadamard product (*i.e.*, mask region extraction, $\mathbf{I} \odot \mathbf{M}$ represents the masked image), $sim$ denotes cosine similarity, $\phi_T, \phi_I$ denotes CLIP's text and image encoders, $\mathcal{M}, \mathcal{M}'$ denotes the initial and final mask instance proposals. For each image $\mathbf{I}$, we also use Depth-Anything-V2 small (Yang et al., 2024a) to generate its depth map $\mathbf{D}$. For each text $\mathbf{T}$, we use the LLM to generate its rephrased version $\mathbf{T}'$. More details are provided in Section C.1 of the Appendix.

**Feature Extraction.** For the visual stream, we use two ViT-based image encoders, DINOv2 (Oquab et al., 2023) and CLIP (Radford et al., 2021). The masked images $\mathbf{I} \odot \mathbf{M}$ are fed into CLIP's visual

encoder to extract class embeddings for each mask, producing instance-level Implicit Semantic features $\mathbf{F}_{\text{cls}} \in \mathbb{R}^{N \times D}$, where N is the number of masks. The entire image is processed by DINO v2 to extract the patch embeddings, which are then reshaped into feature maps $\mathbf{F}_{\text{dino}} \in \mathbb{R}^{H' \times W' \times D}$, where $H'$ and $W'$ denote the patch grid. Using both encoders provides complementary strengths. CLIP, trained on large-scale vision–language data, captures strong category-level semantics aligned with natural language, while DINOv2 preserves fine-grained appearance and structural layout information from self-supervised pretraining. For the textual stream, we use a BERT-like text encoder SimCSE (Gao et al., 2022). The original and rephrased texts $(\mathbf{T}, \mathbf{T}')$ are encoded in global text embeddings $\mathbf{t}, \mathbf{t}' \in \mathbb{R}^d$.

## 3.2 Explicit Spatial Enhancement Module (ESEM)

Although the CLIP features capture implicit semantics of each instance through direct encoding of the masked image, they fail to preserve the geometric context and cannot model inter-instance spatial relations. For example, it is hard to distinguish *"man on the left"* from *"man on the right"* through the naive approach of contrastive learning, since *"left"* and *"right"* semantics are symmetric in the linguistic representation space and cannot be well-comprehended in the absence of geometric information from the visual embeddings.

To address this, we propose the **Explicit Spatial Enhancement Module (ESEM)**, which enriches instance features with explicit geometric information, forming *Spatial Embedding*, as shown in Figure 2. The process has three main steps. **First**, we extract pooled appearance embeddings from the DINO feature maps $\mathbf{F}_{\text{dino}} \in \mathbb{R}^{H' \times W' \times D}$. Each downsampled mask is applied to this feature map, and a mask-guided max pooling is performed to obtain a feature matrix $\mathbf{F}_{\text{pool}} \in \mathbb{R}^{N \times D}$ that captures the most salient feature responses for each of $N$ candidates.

**Second**, we compute geometric properties for each mask. Given the mask candidate $\mathbf{M}_i$ and its downsampled counterpart $M_i'$, we estimate its centroid in the 2D image plane $(x_i, y_i)$ together with the mean depth value $z_i$ from the depth map $\mathbf{D}$ as follows:

$$x_i = \frac{\sum_{h,w} w \cdot \mathbf{M}_i'(h, w)}{\sum_{h,w} \mathbf{M}_i'(h, w)}, \quad y_i = \frac{\sum_{h,w} h \cdot \mathbf{M}_i'(h, w)}{\sum_{h,w} \mathbf{M}_i'(h, w)}, \quad z_i = \frac{\sum_{h,w} \mathbf{D}(h, w) \cdot \mathbf{M}_i(h, w)}{\sum_{h,w} \mathbf{M}_i(h, w)} \quad (2)$$

The resulting triplet $(x_i, y_i, z_i)$ encodes the candidate's position and relative distance in the scene, and $z_i$ is normalized to match the scale of $x_i$ and $y_i$. Notice that the depth retrieval should use the original mask since the pseudo depth provided by the depth estimator is dense.

**Finally**, the pooled appearance features $\mathbf{F}_{\text{pool}} \in \mathbb{R}^{N \times D}$ and the geometric descriptors $\mathbf{G} \in \mathbb{R}^{N \times 3}$ are concatenated into a single feature matrix, which is projected linearly to produce the final spatial embeddings $\mathbf{F}_{\text{geo}} \in \mathbb{R}^{N \times D}$. These embeddings explicitly integrate appearance, position, and depth cues, allowing the model to better capture spatial relationships and resolve cases that cannot be disambiguated by semantics alone.

## 3.3 Weakly Supervised Region-Text Alignment

We establish region-text correspondence under weak supervision through designs in vision, language, and contrastive learning.

**Dual-Similarities Measurement (DualSimFuse).** Each mask candidate is represented by two complementary embeddings: the CLIP class embedding $\mathbf{F}_{\text{cls}} \in \mathbb{R}^{N \times D}$, capturing semantic distinctiveness, and the spatial embedding $\mathbf{F}_{\text{geo}} \in \mathbb{R}^{N \times D}$, enriched by the proposed ESEM with pooled appearance and geometric cues. We jointly measure the similarities between the textual embedding and these two visual embeddings as follows:

$$s = (\mathbf{F}_{\text{cls}})^\top \mathbf{t}^{[1:d]} + \frac{d}{D} \cdot (\mathbf{F}_{\text{geo}})^\top \mathbf{t} \quad (3)$$

where $\mathbf{t}^{[1:d]} \in \mathbb{R}^d$ denotes the first $d$ dimensions of the text embedding $\mathbf{t} \in \mathbb{R}^D$; $(\cdot)^\top$ denotes vector transpose, representing the inner product operation.

**Mask-based Contrastive Learning.** Motivated by the anchor-based contrastive learning in RefCLIP (Jin et al., 2023), we propose a mask-based cross-modal contrastive learning mechanism.

Given the batch size $B$, for each referring text, we calculate its dual-similarity with all the candidates in the batch. Within the same image, we select the candidate that has the maximum similarity with that referring text as the only positive sample, and the rest of the $N - 1$ instances within the image as the negative samples. To further expand the number of the negative samples, we additionally consider all of $N$ candidates within the other $B - 1$ images as the negative samples, adding up to $B \cdot N - 1$ negative samples in total. Therefore, we define the contrastive loss as:

$$\mathcal{L}_{\text{CL}} = -\frac{1}{B} \sum_{i=1}^{B} \log \frac{\exp\left(s_{i,j^+}/\tau\right)}{\exp\left(s_{i,j^+}/\tau\right) + \sum_{k=1}^{B \cdot N - 1} \exp\left(s_{i,k}^-/\tau\right)} \tag{4}$$

where $s_{i,k}^-$ denotes the similarity between the i-th referring text in the batch and its corresponding $k$-th negative instance, and $\tau$ is a temperature hyperparameter.

**Language Consistency Module:** In the Masked-based Contrastive Learning, texts serve as an anchor that drives alignment between the text and visual instances. However, natural language exhibits rich variations in expression. For instance, the referring expression *"person holding a cup"* could be rephrased as *"the one who is grasping a cup"* or *"individual with a cup in hand"*, all describing the same entity despite syntactic differences. Conversely, a subtle change like *"cup held by a person"* may refer to the *"cup"*, not the *"person"*, demanding the model to understand not just structure but also semantic roles and real-world context. To enhance the model's robustness to linguistic variation, we leverage Large Language Models (LLMs) to generate semantically equivalent yet syntactically diverse expressions of the original referring texts. We enforce the model's instance selection distribution to remain consistent between the original and rephrased expressions, encouraging the visual grounding to be invariant to surface form and anchored in deeper semantics. This process is formulated as:

$$\mathcal{L}_{\text{LC}} = \frac{1}{B} \sum_{i=1}^{B} \text{CrossEntropy}\left(\text{SG}(\mathbf{p}_i) \,\|\, \mathbf{p}_i'\right), \tag{5}$$

where $\mathbf{p}_i = \text{softmax}\left(\{s_{i,j}\}_{j=1}^{N}/\tau\right)$ and $\mathbf{p}_i' = \text{softmax}\left(\{s_{i,j}'\}_{j=1}^{N}/\tau\right)$ are the instance selection probability distributions computed using the original text embedding $\mathbf{t}_i$ and its LLM-rephrased counterpart $\mathbf{t}_i'$, respectively. $\text{SG}()$ denotes stop gradient. $s_{i,j}$ and $s_{i,j}'$ are computed following Eq. 3.

**Training and Inference.** The total loss function is $\mathcal{L} = \mathcal{L}_{\text{CL}} + \mathcal{L}_{\text{LC}}$. For inference, the mask candidate that has the highest similarity with the original referring text is selected as the prediction.

### 3.4 CONFIDENCE-AWARE DENSE DISTILLATION (CADD)

To overcome the limitations of relying solely on SAM proposals, and to enable efficient inference with a more lightweight model which can distill knowledge from the previous alignment model, rather than those heavy pretrained foundation models, we introduce a Confidence-Aware Dense Distillation (CADD) strategy. We use the predicted masks from previous steps to train a lightweight referring segmentation network, *e.g.,* LAVT (Yang et al., 2022). We adopt student-teacher network training strategies. Details about network architectures and training strategies are provided in Section C.3 of the Appendix.

## 4 EXPERIMENTAL RESULTS

### 4.1 EXPERIMENTAL SETTINGS

**Datasets.** We evaluate the proposed method on three datasets: RefCOCO Yu et al. (2016), RefCOCO+ Yu et al. (2016), and Gref Mao et al. (2016). These three datasets were built based on MSCOCO Lin et al. (2015), consisting of images with annotated natural language expressions. RefCOCO contains 142,209 refer expressions for 50,000 objects across 19,994 images, while RefCOCO+ includes 141,564 expressions for 49,856 objects in 19,992 images. Gref dataset includes 104,560 expressions for 54,822 objects in 26,711 images. RefCOCO expressions primarily focus on the spatial positioning of objects, whereas RefCOCO+ expressions emphasize object appearance. Gref expressions present a greater challenge as they are generally longer and more complex.

| Methods | Sup. | Extra Training Data | RefCOCO | | | RefCOCO+ | | | Gref |
|---|---|---|---|---|---|---|---|---|---|
| | | | val | testA | testB | val | testA | testB | val |
| Strudel *et al.* (Arxiv) Strudel et al. (2022) | MT | ✓ | 25.44 | - | - | 22.01 | - | - | 22.05 |
| Liu *et al.* (ICCV) Liu et al. (2023) | MT | ✗ | 31.17 | 32.43 | 29.56 | 30.90 | 30.42 | 30.80 | 36.00 |
| PCNet (NeurIPS) Yang et al. (2024b) | MT | ✗ | 52.20 | 58.40 | 42.10 | 47.90 | 56.50 | 36.20 | 47.30 |
| ES³Net(*Ours*) | MT | ✗ | **67.38** | **72.18** | **59.88** | **55.54** | **64.37** | **42.50** | **55.04** |
| Group-ViT (CVPR) Xu et al. (2022a) | ST | ✓ | 12.97 | 14.98 | 12.02 | 13.21 | 15.08 | 12.41 | 16.84 |
| Lee *et al.* (ICCV) Lee et al. (2023) | ST | ✗ | 31.06 | 32.30 | 30.11 | 31.28 | 32.11 | 30.13 | 32.88 |
| SaG (ICCV) Kim et al. (2023) | ST | ✗ | 34.76 | 34.58 | 35.01 | 28.48 | 28.60 | 27.98 | 28.87 |
| PPT (CVPR) Dai & Yang (2024) | ST | ✓ | 46.76 | 45.33 | 46.28 | 45.34 | 45.84 | **44.77** | 42.97 |
| WeakMCN (CVPR) Cheng et al. (2025) | ST | ✗ | 59.46 | 61.01 | 56.40 | 44.36 | 50.40 | 37.12 | 46.81 |
| ES³Net(Ours) | ST | ✗ | **63.80** | **68.49** | **58.71** | **51.89** | **61.30** | 40.27 | **52.27** |
| Our Oracle (LAVT) | F | ✗ | 72.73 | 75.82 | 68.79 | 62.14 | 68.38 | 55.10 | 60.50 |

Table 1: Comparison with the state-of-the-art RIS methods with different levels of supervision (MT: supervised by multiple texts referring one image, with knowledge whether any two texts refer to the same instance or not. ST: supervised by single text paired with one image, and no explicit knowledge of relations between texts is given. F: supervised by ground-truth labels) in mIoU (%). The best and the second best results are **bold** and underlined.

| ESEM | LCM | RefCOCO | RefCOCO+ | G-Ref |
|---|---|---|---|---|
| | | val / testA / testB | val / testA / testB | val |
| | | 45.76 / 52.08 / 37.20 | 45.76 / 54.59 / 35.41 | 47.30 |
| ✓ | | 58.18 / 61.44 / 53.86 | 48.63 / 56.96 / 39.85 | 49.99 |
| ✓ | ✓ | **59.26 / 64.14 / 54.43** | **49.37 / 57.62 / 40.66** | **51.64** |

Table 2: Ablation study on the proposed Explicit Spatial Enhancement Module (ESEM) and Language Consistency Module (LCM). The experiments are all conducted without Confidence-Aware Dense Distillation (CADD).

**Evaluation metric.** Following the common practice Kim et al. (2023); Lee et al. (2023); Liu et al. (2023), we use the mean intersection-over-union (mIoU) to evaluate performance.

**Implementation details.** We adopt SAM2-Small (Ravi et al., 2024) to generate candidate masks, Depth-Anything-V2-Small (Yang et al., 2024a) to produce depth maps, and DeepSeek-V3 (via API) (DeepSeek-AI, 2024) to generate semantically equivalent rephrasings of the referring texts. Candidate instances are coarsely ranked and filtered using CLIP-ViT-Base16 (Radford et al., 2021). By default, the network is trained on 2 NVIDIA RTX 3090 GPUs with a total batch size of 200, optimized via AdamW (Loshchilov & Hutter, 2019), where the learning rate is set to 0.0005 and the weight decay to 0.0. The number of candidate instances per image, $N$, is set at 10 during training and 25 for evaluation. We used pre-trained and frozen CLIP-ViT-Base16 to extract the [CLS] token and DINOv2-Base (Oquab et al., 2023) to extract patch tokens. The temperature $\tau$ is set to 1.0 for both $\mathcal{L}_{CL}$ and $\mathcal{L}_{LC}$. For the CADD module, training details are provided in the Appendix C.3.

### 4.2 COMPARISON TO STATE-OF-THE-ARTS

Table 1 compares our proposed method with various *SoTA* methods in WSRIS, which are grouped into two categories: MT (Multi-text supervision) and ST (Single-text supervision). In the ST domain, which relies solely on image-text pairs supervision, our method surpasses WeakMCN (Cheng et al., 2025) on all dataset at a significant margin. For instance, on the challenging dataset Ref-COCO+, which contains more complicated, descriptive referring texts and inter-object relationship than RefCOCO, our method achieves a performance boost of over 17.0%(+7.53), 21.6%(+10.90) and 8.5%(+3.15) on the *val, testA and testB* set compared to WeakMCN. In the MT domain, the results demonstrate the efficacy of our method's adaptability under MT supervision and strong capability compared to the MT-based WSRIS methods and even the fully supervised method. For example, on the RefCOCO dataset, our method surpasses PCNet (Yang et al., 2024b) by 29.1%(+15.18), 23.6%(+13.78) and 42.2%(+17.78) on *val, testA* and *testB* set. It should be noticed that on the *testA* set, there is only a 3.64 performance gap between ours and LAVT (Yang et al., 2022), our oracle

*Q*: *the white thing next to the mouse on the right side*

*Q*: *black hat shown from the back in the front of the picture*

*Q*: *the woman with the umbrella and her hand on her face at the right side*

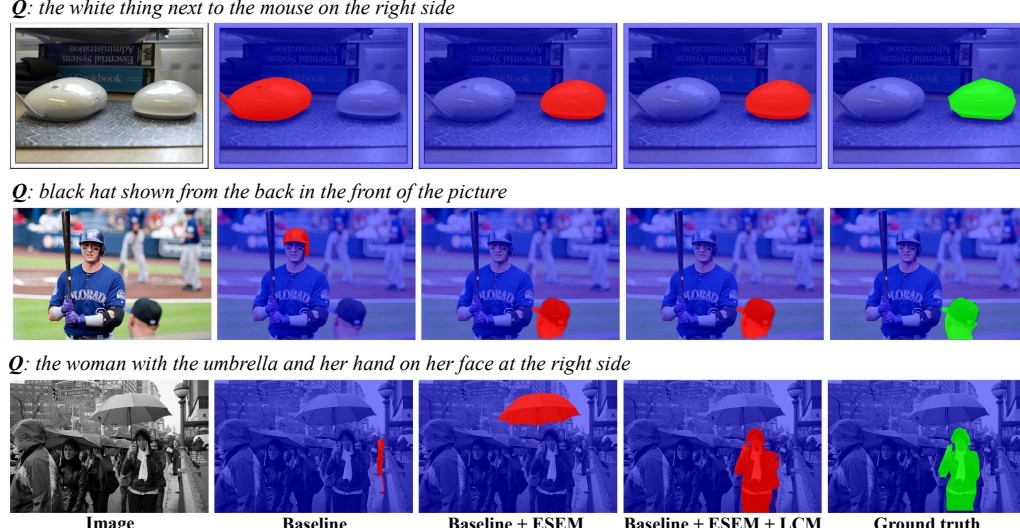

| Image | Baseline | Baseline + ESEM | Baseline + ESEM + LCM | Ground truth |

Figure 3: Visualization of the ablation studies to demonstrate the effectiveness of each component. In the first and second row, it could be visualized that the ESEM successfully captures the spatial related cues. In the third row, it could be visualized that the LCM correctly help the model recognize the subject of the sentence and correct the prediction.

| Patch Tokens | Class Tokens | RefCOCO
*val / testA / testB* | RefCOCO+
*val / testA / testB* | G-Ref
*val* |
|---|---|---|---|---|
| CLIP | DINOv2 | 34.70 / 36.45 / 32.54 | 33.10 / 36.05 / 29.36 | 38.94 |
| CLIP | CLIP | 48.47 / 57.15 / 39.85 | 46.95 / 56.47 / 36.04 | 47.78 |
| DINOv2 | DINOv2 | 51.82 / 56.89 / 47.22 | 43.70 / 50.11 / 37.01 | 50.66 |
| DINOv2 | CLIP | **59.26 / 64.14 / 54.43** | **49.37 / 57.62 / 40.66** | **51.64** |

Table 3: Ablation study on the vision encoders usage for semantic-spatial alignment in ESEM. LCM is enabled while CADD is deactivated.

model, which is trained with ground-truth labels. More details about the difference between MT and ST, and our MT implementation are provided in the Section B.

### 4.3 ABLATION STUDY

**Components Analysis.** We evaluated the result of our baseline, which solely relies on aligning CLIP's visual class tokens with the text encoder through projection layers. CLIP is capable of capturing object semantics, yet it fails to understand spatial relationships among multiple instances. As illustrated in Table 2, adding ESEM significantly improves performance on all datasets, and it is worth noting that we achieve substantial gains of 27.1%(+12.42), 18.0%(+9.36) and 44.8%(+16.66) on RefCOCO *val, testA* and *testB* set, which contains large amounts of referring text with respect to spatial understanding. The first two rows in Figure 3 also indicate a significant improvement in comprehending spatial keywords such as *"on the right side"* and *"in the front of"*. Adding LCM further improves the performance on three datasets, realizing the greatest gain on the RefCOCO *testA* set of 4.4%(+2.7). As illustrated in the last row in Figure 3, LCM enhances model's robustness in handling complex sentences to identify the real referring target. More qualitative results regarding ESEM and LCM are visualized in Section E.

**Analysis of Vision Encoders.** We analyze the performance of different vision encoders in our framework, focusing on the choice of patch and CLS tokens. Table 3 shows that the combination of DINOv2 patch tokens with CLIP CLS token achieves the best results across all benchmarks, significantly outperforming other configurations. Notably, CLIP's CLS token consistently yields higher scores than DINOv2's, indicating its inherent instance semantic representation capability for reference grounding tasks. DINOv2's patch tokens provide richer spatial details, contributing to improved localization accuracy. This synergy demonstrates that leveraging complementary strengths

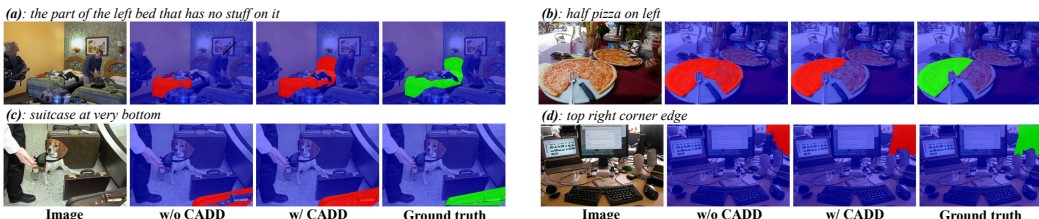

Figure 4: Visualization of the ablation study on using (w/) and not using (w/o) the CADD strategy. The proposed CADD module successfully produces a better mask prediction than even the best mask candidates proposed by SAM.

from DINOv2 for spatial visual cues and CLIP for instance semantic represents an optimal design choice for vision encoders in our architecture.

**Analysis of Spatial Encoding.** We analyze the efficacy of some technical details in our ESEM. Since we are evaluating model's spatial comprehension capability, we collect all the samples involving spatial context on RefCOCO, RefCOCO+ and G-Ref, merge them into one spatial-related dataset and evaluate on it. As illustrated in the Table 4 first two rows, the result demonstrates the necessity of the channel-wise max-pooling method compared to the vanilla mean-pooling method, which might be skilled at capturing the most significant feature over the masked region. The last two rows show that simply simply adding 2D centroid information and further 3D centroid information could yield a performance gain of 8.22%(+3.9) and 3.6%(+1.86) respectively.

| Pooling | Spatial Info. | mIoU(%) |
|---------|---------------|---------|
| mean | None | 45.82 |
| max | None | 47.44 |
| max | 2D | 51.34 |
| max | 3D | **53.20** |

Table 4: Effect of Different Spatial Encoding Strategies. 2D and 3D centroids provides explicit spatial information, bringing considerable performance gain.

**Analysis of the CADD Strategy.** We investigate the effectiveness of the Confidence-Aware Dense Distillation (CADD) module, which distills fine-grained segmentation knowledge from ES$^3$Net's coarse selection result to refine and denoise its prediction. As shown in Table 5, CADD yields consistent improvements in all datasets, with significant gains in RefCOCO. This is because off-the-shelf proposal generators like SAM are semantic-agnostic and often

| CADD | RefCOCO _val / testA /testB_ | RefCOCO+ _val / testA / testB_ | G-Ref _val_ |
|------|------------------------------|--------------------------------|-------------|
| | 59.26 / 64.14 / 54.43 | 49.37 / 57.62 / **40.66** | 51.64 |
| ✓ | **63.80 / 68.49 / 58.71** | **51.89 / 61.30** / 40.27 | **52.27** |

Table 5: Effect of CADD. Overall, the CADD module successfully distills a segmentor that provides better prediction than solely selecting SAM proposals, overcoming the problem that the SAM proposal mismatches with the ground truth.

fail to produce accurate masks for fine-grained or spatially complex expressions (*e.g., "part of bed"* or *"half pizza on the left"*). In such cases, ES$^3$Net itself cannot select a suitable candidate from the proposals. CADD overcomes this limitation by learning a dedicated segmentation head guided by dense, confidence-aware supervision from ES$^3$Net selection, which bypasses SAM's proposal constraints entirely. The qualitative results in Figure 4 further validate that CADD produces masks more accurate and coherent than even the best SAM proposals. More qualitative visualization results are shown in Section E.

## 5 CONCLUSION

In this work, we addressed the fundamental limitation of WSRIS, the lack of explicit semantic–spatial synergy, and introduced ES$^3$Net, a novel framework to overcome this gap. More specifically, we propose the Explicit Spatial Enhancement Module which injects geometric context to strengthen spatial awareness, the Language Consistency Module which improves robustness under diverse expressions, and the Confidence-Aware Dense Distillation strategy which stabilizes learning and anables efficient inference. Through this unified design, the proposed ES$^3$Net significantly narrows the gap between weakly and fully supervised methods and establishes new state-of-the-art WSRIS performance on RefCOCO, RefCOCO+, and Gref. These results demonstrate that explicitly modeling semantic–spatial synergy is crucial for advancing WSRIS.

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

## A  THE USE OF LARGE LANGUAGE MODELS (LLMS)

LLMs are only used to help polish paper writing. The retrieval of references and ideation of research are performed solely by human authors.

## B  ARGUMENT ON SINGLE-TEXT AND MULTI-TEXT SUPERVISION

Multi-text prior (MT), which additionally groups referring expressions by the object they describe in the same image, is naturally available in curated datasets like RefCOCO/+/G-Ref, where multiple captions are intentionally written for the same region. While some methods exploit this for better instance grounding, in real-world web-scale image-text collections, such structured grouping is almost never available: captions are often independently written, noisy, or refer to different parts of the image without annotation alignment. Obtaining MT supervision from the web would require costly human verification, which is impractical at scale.

The MT prior in fact brings more information and stronger supervision. For our model, the performance improvement can be in fact verified from a mathematical perspective. There is in fact no extra training needed. The MT can be forced so that the proposal selection quality is improved thus benefiting the CADD distilled segmentor.

**Preliminary:** Through statistics, we found our model's proposal selection correctness and it's confidence of selection shows linear relation as shown in Figure 5. The correlation coefficient $R$ is up to 0.986, which is almost to the limit 1. Then we may formulate the probability of selection correctness from the i-th prediction as:

$$P_i = k \cdot C_i + b \tag{6}$$

where $C_i$ denotes the confidence of the i-th prediction.

**An mathematically verified way to improve proposal selection via MT prior:** Suppose there are $n$ textual cues that refer to instances within one image. According to the MT prior, we may assign the textual cues into $m$ groups, i.e., $G_1, G_2, ..., G_m$. All cues in the same group are constrained to make the same proposal selection. We claim that the strategy that each cue in the same group makes the same selection as the the most confident cue in the group will yield better result:

Proof: The probability of correctness of the most confident cue is $P_t = k \cdot \max(C_i \in G_t) + b, t \in \{1, 2, ..., m\}$ according to Equation 6. Thus the expectation number of correctness cases would be $E_{corrected} = \Sigma_{t=1}^m |G_m| \cdot (k \cdot \max(C_i \in G_t) + b) = n \cdot b + k \cdot \Sigma_{t=1}^m |G_m| \cdot \max(C_i \in G_t)$. Since $\max(C_i \in G_t) \geq C_j, \forall C_j \in G_t$, we have $E_{corrected} = n \cdot b + k \cdot \Sigma_{t=1}^m |G_m| \cdot \max(C_i \in G_t) \geq n \cdot b + k \cdot \Sigma_{i=1}^n C_i = \Sigma_{i=1}^n (k \cdot C_i + b) = \Sigma_{i=1}^n P_i = E_{uncorrected}$, where $E_{uncorrected}$ denote the expectation without MT prior correction. Thus, we verify $E_{corrected} \geq E_{uncorrected}$. In fact, this conclusion can be extended to scenarios where confidence is positively correlated with correctness, not necessarily linear.

In practice, we also know the fact that cues in different groups should select different proposals, thus we propose a greedy matching algorithm to fully utilize the MT prior. The algorithm first calculates the total confidence in a group, i.e. for group $G_t$, $C_t^{total} = \Sigma C_i, C_i \in G_t$. The group with the greatest total confidence first selects a proposal corresponding to the image. Then the 2nd group with 2nd largest total confidence select a proposal. It should noted if the selected proposal overlaps, the latter group can only choose another proposal that is most favored within the proposals not yet choosen.

Extensive experiments in Table 6 show that greedy matching is a even better correction method for the selected proposals, largely improves the label supervision for CADD module. It also worth noticing that the MT prior is only used in training. Inference need no MT prior following previous works. Through utilizing MT prior with greedy matching in training, we yield quite promising result as shown in Table 1.

In Weakly Supervised Referring Image Segmentation (WSRIS), explicitly distinguishing ST (Single-Text) and MT (Multi-Text) supervision is therefore essential not just for fair evaluation, but to reflect what's realistically obtainable from internet-scale data, and to guide research toward methods that work under true weak supervision.

| Methods | RefCOCO | RefCOCO+ | G-Ref |
|---|---|---|---|
| ST | 56.88 | 49.85 | 51.67 |
| Proved method | 59.98 | 51.38 | 53.04 |
| Greedy matching | **63.18** | **54.14** | **54.82** |

Table 6: The comparison of three methods' mask selection quality before providing label supervision to CADD, measured by mIoU. The proved method is the method with a formal mathematical proof, while greedy matching yields the best result. All results are reported from train split since MT prior is not used during inference time.

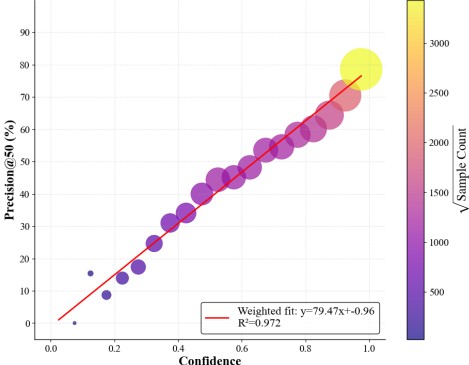

Figure 5: Our model's prediction correctness probability (measured by Pr@50) is positively linearly correlated to the confidence of its prediction. Statistics are from RefCOCO. Other datasets also present linear relations.

---

You are a professional data augmentation assistant specialized in generating training text for the **Referring Image Segmentation** task.

## Task

For each of the following **<n>** referring expressions, generate exactly **<k>** augmented versions.
You **must**:
- Preserve the core semantics and the referred object exactly
- Maintain all spatial relationships (e.g., left, right, top, center) **unchanged**
- Use synonym substitution only when it **does not alter the meaning**
- Imitate the natural, diverse, and colloquial style observed in human-annotated datasets
- If the input contains grammatical errors or obvious typos, first correct them,
then perform augmentation based on the corrected version

## Diversity Requirements

Maximize variation across the following dimensions while preserving referential equivalence:
- **Ellipsis**: e.g., using vague determiners like "a certain...", "the one..."
- **Redundancy**: e.g., adding harmless modifiers such as "the ... that is...",
"the ... which appears to be..."
- **Perspective shift**: e.g., "that ..." v.s. "the ... located at..."
- **Syntactic reordering**: e.g., fronting adverbials, postposing modifiers
- **Lexical substitution**: e.g., "standing" v.s "situated", "near" v.s. "close to"
- **Sentence structure variation**: simple v.s complex sentences, active v.s passive voice

## Output Format
- Return results in a **minimal, valid JSON object**
- Use the key `"a"` for the list of augmented strings
- Output **exactly one JSON object**: `{{"r": [...]}}`
- Generate **precisely <n> entries**, each corresponding to one input expression.

## Input
{input_section}

## Output (must be valid and minimal JSON):
{{"r":[{{"a":["",""]}}]}}.

Figure 6: LLM prompts used for generating rephrased texts in batch. Placeholders including the processed batch size ($n$) and requested number of rephrased texts ($k$) are marked by blue colors.

# C   MORE IMPLEMENTATION DETAILS

## C.1   GENERATION OF REPHRASED TEXTS

To enrich linguistic diversity while preserving referential grounding, we generate paraphrased refer-ring expressions using Deepseek-V3 (DeepSeek-AI, 2024) via API. For each original expression, we produce $k = 3$ rephrased texts that maintain the same referent and spatial relationships (e.g., left, right, center), following the natural and colloquial style of human annotations in RefCOCO (Yu et al., 2016). The augmentation is performed in batches of $N = 10$ expressions per API request to improve throughput while ensuring structured output. Each batch is processed via a single LLM call with the following generation hyper-parameters: (i) $temperature = 0.75$; (ii) $top - p = 0.95$. The prompt explicitly instructs the model to output a minimal JSON object containing exactly $k$ para-phrases for each of the 10 inputs. Invalid or malformed JSON responses are repaired heuristically when possible; otherwise, empty strings are used to retain alignment with the original data. Each sample is saved as an individual JSON file containing the original expression and its $n$ augmented variants, enabling straightforward integration into downstream training pipelines. The prompt is illustrated in Figure 6.

## C.2   MASK PROPOSALS GENERATION PIPELINE

Our method needs pre-generated mask proposals to aid training. However, directly use SAM's everything mode usually yield over-segmented candidates. For example, for a man wearing T-shirt, SAM's everything model often gives arms, T-shirt, head separately, instead of provide mask for the entire man. Moreover, the everything mode is quite costly compared to using single prompts. As a result, in reality, we deploy a detection model Detic Zhou et al. (2022) trained on ImageNet (avoid the possibility of data leakage from COCO dataset) to generate bounding boxes as prompts to SAM. By contrast, the detection model that WeakMCN Cheng et al. (2025) uses a detector trained on COCO dataset which can be unfair. The inferiority of Detic than COCO-trained YOLO on COCO dataset in metrics such as mAP can be verified. Moreover, the class information is discarded since we do not need any information about what is in the bounding box at all. We only want a mask that can in fact represent an instance in the image.

Thus through this Detic+SAM pipeline, we can get quality mask proposals. For each mask, we extract the corresponding area in the image and compare it with the referring text through CLIP encoding to attain a cosine similarity. The final proposals are ranked by this similarity. We explicitly measured the quality of the ranked generated proposals which counts the best IoU between top-N ranked proposals and the ground truth. The final mIoU versus the number of top ranked proposals are shown in Figure 7. As can be noticed, the mIoU at top-10 is already around 80, which is basically satisfactory for training.

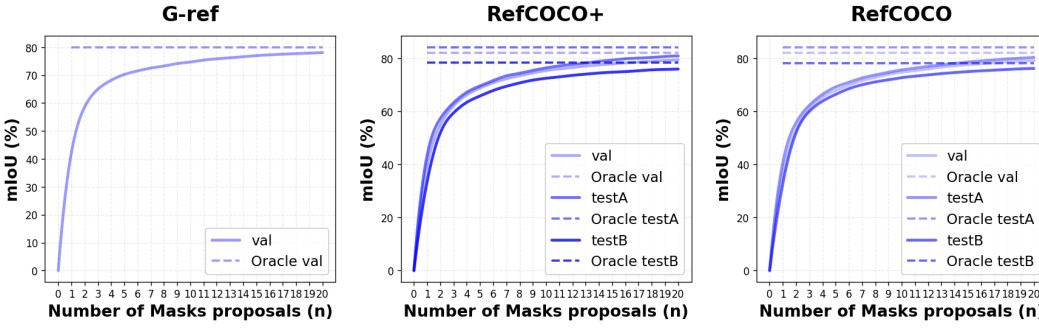

Figure 7: The mIoU ascends as the number of mask proposals increases. This means more proposals that match the ground truth appear more as the number of proposals increase.

## C.3 MORE EXPLANATION ABOUT CADD

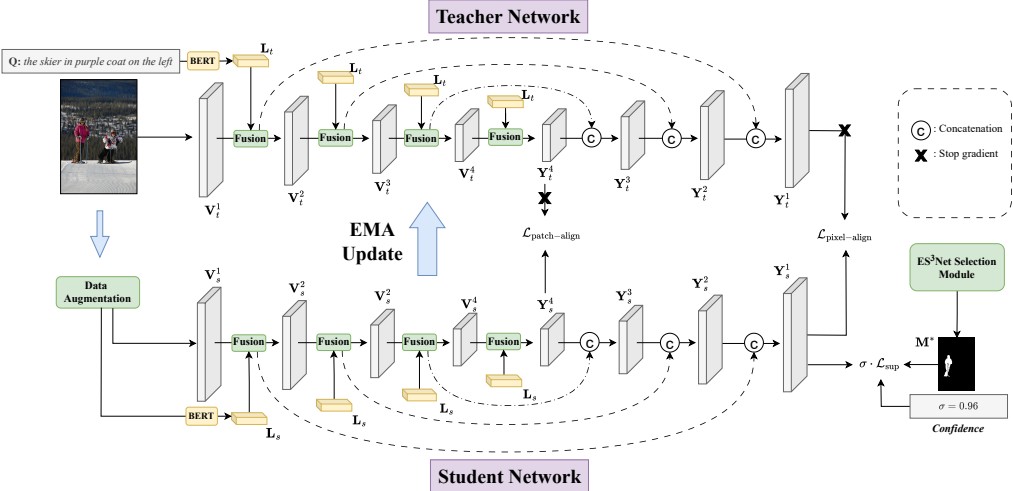

Figure 8: Overview of our CADD component.

We now describe CADD proposed in Section 3.4 with more details. As shown in Figure 8, our framework employs a student-teacher architecture based on LAVT (Yang et al., 2022), a fully supervised Referring Image Segmentation model. The teacher network processes the original image-text pair $(\mathbf{I}, \mathbf{T})$ to provide stable predictions. The student network is trained on augmented data, including LLM-generated referring expressions $\mathbf{T}'$ as well as rule-based transformed images $\mathbf{I}'$, and learns under the guidance of the teacher.

Let $\mathbf{Y}_s^1$ and $\mathbf{Y}_t^1$ denote the final segmentation logits from the student and teacher networks, respectively, and let $\mathbf{Y}_s^4, \mathbf{Y}_t^4$ denote their patch token representations at the output of the final cross-modal transformer layer. The student network is trained with three loss terms. **First**, a confidence-weighted supervised loss:

$$\mathcal{L}_{\text{sup}} = \sigma \cdot \text{CrossEntropy}(\mathbf{Y}_s^1, \mathbf{M}^*), \tag{7}$$

where $\mathbf{M}^*$ is the mask candidate selected by the previous ES$^3$Net modules; $\sigma \in [0, 1]$ is the confidence score associated with $\mathbf{M}^*$, and CrossEntropy denotes the pixel-wise cross-entropy loss. **Second**, a pixel-level consistency loss:

$$\mathcal{L}_{\text{pixel-align}} = \text{SoftDiceLoss}(\mathbf{Y}_s^1, \mathbf{Y}_t^1), \tag{8}$$

where SoftDiceLoss denotes the soft dice loss between the student's and teacher's final predictions. **Third**, a patch-level consistency loss that explicitly enforces agreement between deep features in the last visual-language encoder layer:

$$\mathcal{L}_{\text{patch-align}} = \text{MSE}(\mathbf{Y}_s^4, \mathbf{Y}_t^4). \tag{9}$$

where MSE denotes the mean square error.

In summary, the total loss for the student is:

$$\mathcal{L} = \mathcal{L}_{\text{sup}} + \lambda_1 \mathcal{L}_{\text{pixel-align}} + \lambda_2 \mathcal{L}_{\text{patch-align}}, \tag{10}$$

with $\lambda_1 = 0.5, \lambda_2 = 0.05$ in our experiments.

The teacher parameters $\theta_t$ are updated via exponential moving average (EMA) (Tarvainen & Valpola, 2018) of the student parameters $\theta_s$:

$$\theta_t \leftarrow \alpha \theta_t + (1 - \alpha)\theta_s, \tag{11}$$

where $\alpha = 0.9996$. Gradients are not back-propagated through the teacher network, ensuring stable supervision. This formulation allows the student to benefit from diverse linguistic augmentations while being regularized by the teacher's consistent predictions at both output and deep feature levels.

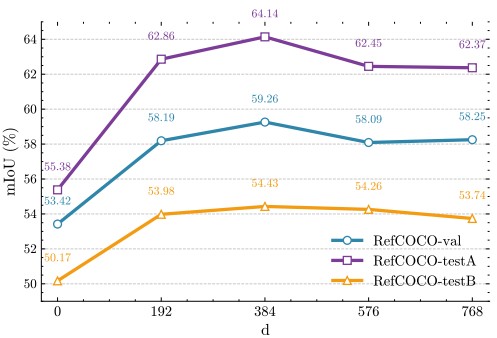 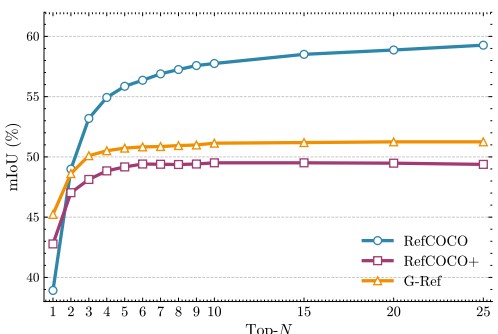

Figure 9: Effect of the instance feature dimension $d$ in the Dual-Similarity Measurement stage on RefCOCO (*val / testA / testB*). The spatial feature dimension $D$ is fixed to 768 during training. Specially, when $d = 0$, we simply control the feature ratio in Equation 3 to $r = \frac{d}{D} = 1$, where we only consider the ESEM feature for alignment.

Figure 10: Impact of $N$ (number of mask candidates) on mIoU% during evaluation across RefCOCO, RefCOCO+, and G-Ref *val* sets. All models are pretrained on the corresponding datasets with fixed $N = 10$.

Through these gradual and progressive distillation procedures, our CADD module would ultimately assimilate knowledge from the previous sophisticated selection modules and gradually optimize its own referring segmentation capability.

**Training Details.**    The training is conducted on 4 NVIDIA RTX 3090 GPUs with a total batch size of 48. Following the strategy of Sun et al. (2023), we begin with a 4-epoch warm-up phase, during which a single student network is trained exclusively using labels provided by the ES³Net selection module. This stage equips the student with an initial capability for referring segmentation. Upon completion of warm-up, the teacher network is initialized by copying the student's parameters. The full student-teacher framework is then trained for 10 additional epochs with a constant learning rate of 0.00005. During this stage, the teacher receives only the original image-text pairs without any augmentation, whereas the student's inputs are subjected to random geometric transformations (*e.g.*, flipping and random cropping) and mild color jitter. All remaining hyper-parameters are kept identical to those used in the oracle setting.

# D    MORE QUANTITATIVE STUDIES

## D.1    ANALYSIS OF FEATURE DIMENSION IN DUALSIMFUSE

We study the effect of the CLIP feature dimension $d$ used for instance-level semantic alignment in the Dual-Similarity Measurement stage. We fix the ESSM spatial feature's dimension to $D = 768$ in all experiments. As shown in Figure 9, performance on RefCOCO peaks at $d = 384$ , indicating an optimal trade-off between semantic richness and spatial grounding. When $d$ is too small (*e.g.*, 192), the model lacks sufficient semantic discriminability to distinguish fine-grained referring expressions. Conversely, larger dimensions (*e.g.*, 768) over-emphasize CLIP-based semantics, diluting the contribution of spatial context from ESEM and leading to degraded performance, which is particularly on *testB* that contains more spatially complex queries. The choice of $d = 384$ thus strikes a balanced integration of instance-level semantics and spatial comprehension, enabling robust alignment without overwhelming the spatial cues critical for accurate segmentation.

## D.2    ANALYSIS OF THE PARAMETER TOP-$N$

To further investigate our network's generalization capability, we train all models with a fixed set of $N = 10$ mask candidates per image that is selected by CLIP-based pre-scoring. During evaluation, however, we expand $N$ to include additional candidates, even those initially ranked low by CLIP. As visualized in Fig. 10, remarkably, segmentation performance consistently improves as $N$ increases across RefCOCO. This demonstrates that our network has learned transferable spatial-

semantic comprehension rather than overfitting to high-scoring proposals selected by frozen-CLIP. It can effectively re-rank low-confidence mask, validating its robustness to candidate quality. This property enables efficient training on sparse candidate sets while supporting high-recall inference, which is a practical advantage for real-world referring segmentation where exhaustive proposal generation is costly. However, we observe no significant improvement on RefCOCO+ and G-Ref. We attribute this to the fact that these datasets contain fewer spatially grounded expressions and instead feature more appearance-based or descriptive referring phrases (*e.g., "the red cup"* or *"a striped cat"*), which align well with CLIP's pre-trained semantic knowledge and thus benefit less from explicit spatial modeling.

## E    MORE QUALITATIVE VISUALIZATIONS

To further illustrate the effectiveness of our proposed modules, we provide additional qualitative visualizations: for ESSM in Section 3.2 (Figure 11), for LCM in Section 3.3 (Figure 12), and for CADD in Section 3.4 (Figure 13).

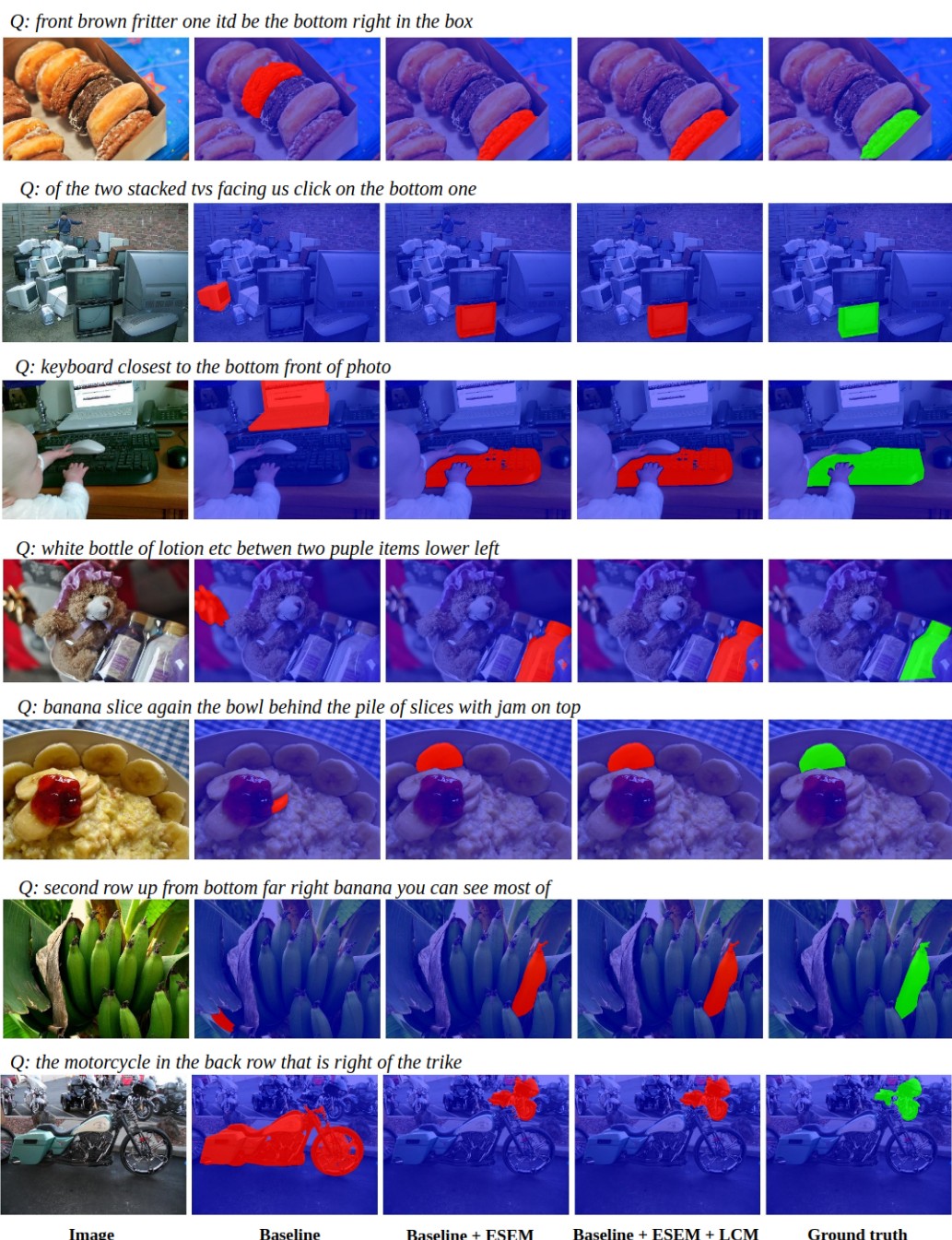

Figure 11: More visualization results of ESEM, which significantly enhance network's spatial understanding capability.

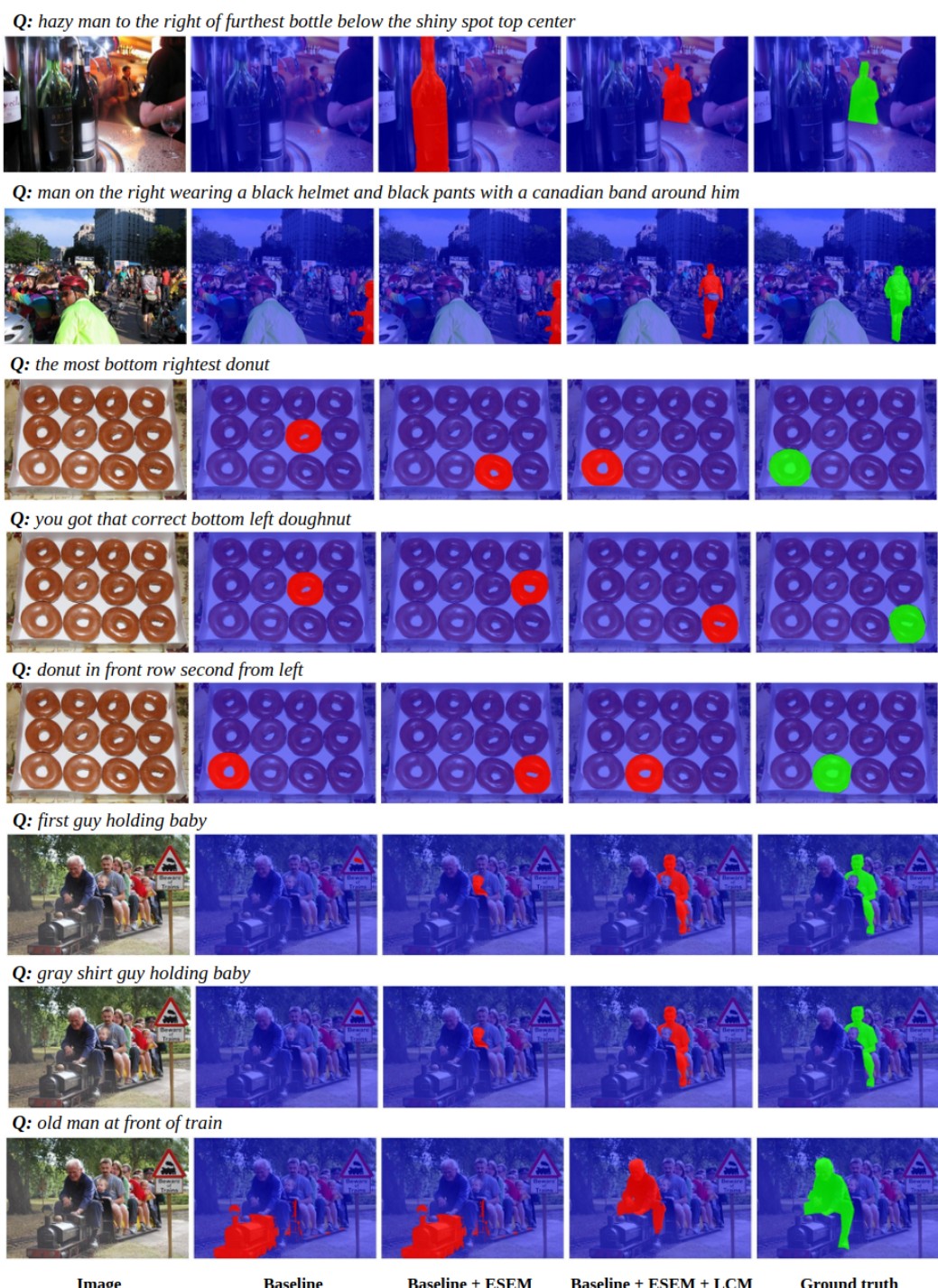

*Q: hazy man to the right of furthest bottle below the shiny spot top center*

*Q: man on the right wearing a black helmet and black pants with a canadian band around him*

*Q: the most bottom rightest donut*

*Q: you got that correct bottom left doughnut*

*Q: donut in front row second from left*

*Q: first guy holding baby*

*Q: gray shirt guy holding baby*

*Q: old man at front of train*

| Image | Baseline | Baseline + ESEM | Baseline + ESEM + LCM | Ground truth |

Figure 12: More visualization results of LCM, which improves model's capability in understanding complex sentences and distinguish the referring targets.

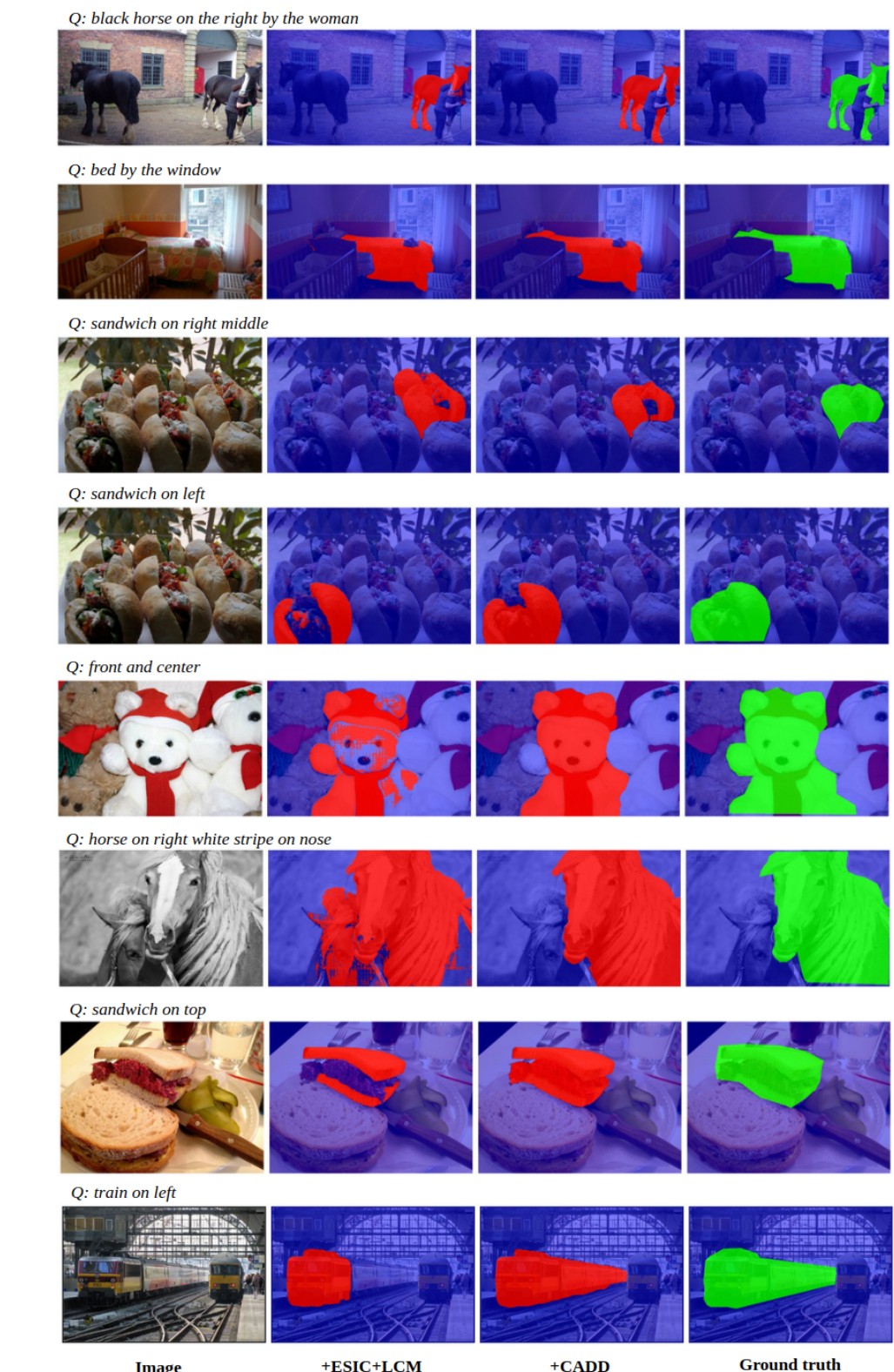

Figure 13: More visualization results of CADD, which learns to predict final semantic-aware fine-grained mask.

