# OpenReview forum: "Learning Explicit Semantic-Spatial Synergy for Weakly Supervised Referring Image Segmentation"
_ICLR.cc/2026/Conference — ICLR 2026 Conference Withdrawn Submission_

### Official Review · Reviewer_ySRT · 2025-10-30

**Soundness:** 2
**Presentation:** 3
**Contribution:** 2
**Rating:** 6
**Confidence:** 5

**Summary:**

Recent advances have improved semantic grounding, existing methods still lack explicit mechanisms to incorporate spatial understanding, preventing them from achieving semantic–spatial synergy.
To address this limitation, it proposes ES3Net framework to explicitly learn Semantic-Spatial Synergy by introducing visual foundation models.

**Strengths:**

- This work proposes ES3Net  framework to explicitly learn Semantic-Spatial Synergy by introducing visual foundation models DepthAnything.

- Extensive experiments on RefCOCO, RefCOCO+, and RefCOCOg demonstrate that ES3Net establishes new state-of-the-art performance

- The overall writing is good and easy for reading.

**Weaknesses:**

- Some concerns about method design
  - It is confused that the Eq.(3) mentions that $t[1:d] \in R^d$ denotes the first d dimensions while $F_{cls} \in R^{D}$. How can the inner product be computed?
  - In the Language Consistency Module, is it necessary to use the stop gradient? In my opinion, when using the stop gradient, it seems to be distilling the knowledge from LLM into the proposed framework. If so, how to avoid the negative effect of noisy labels.
  - In line-271, the candidate that has the maximum similarity with referring text is selected as the positive sample, which is similar to the previous PCNet (which selects the proposal based on heat map). My concern is how to evaluate the effect of incorrect pseudo labels, considering that the fused embedding $F_{geo}$ is not well aligned with the text embedding.

- Experiments
  - The training efficiency evaluation is needed considering the introduction of two vision-foundation models.
  - The recent works such as `LocalizationHeads` ~[A] has achieved outstanding performance on RIS task. The authors should clarify the significance of current weakly-supervised works.
  - The author can provide some generated rephrased text examples and failure cases.
  -  It is a more common practice to put the caption of the table above the table.

In my opinion, the primary contribution of this work lies in fully leveraging the pseudo-label and extra spatial features from pre-trained VLM/LMs to supplement and enhance the vanilla contrastive learning. I may adjust my score based on the authors' response.

[A] CVPR2025, Your Large Vision-Language Model Only Needs A Few Attention Heads For Visual Grounding

**Questions:**

Refer to Weaknesses.

---

### Official Review · Reviewer_u7XL · 2025-10-30

**Soundness:** 2
**Presentation:** 2
**Contribution:** 2
**Rating:** 4
**Confidence:** 3

**Summary:**

This paper focuses on weakly supervised referring image segmentation (WSRIS) as its main task. To address the difficulty that existing WSRIS methods face in understanding spatial information, the paper  proposes an explicit spatial enhancement module. This module utilizes mask-grounded semantic features and a dense map to derive 3D coordinates, which are then fed into the segmentation model to improve performance. Experimentally, the proposed method achieves state-of-the-art performance on several popular WSRIS benchmarks.

**Strengths:**

- The explicit spatial enhancement module leverages off-the-shelf pretrained models such as a mask generator and a depth estimator to generate 3D coordinates, and demonstrates that applying these as spatial embeddings to the segmentation model leads to performance improvement.


- The paper also proposes a language consistency module, which utilizes a large language model (e.g. Deepseek) to obtain rephrased text from the ground truth (GT) descriptions and applies a consistency loss among them to enhance the text encoder.


- The proposed method achieves state-of-the-art performance on several popular WSRIS benchmarks.

**Weaknesses:**

- Fairness as a Weakly Supervised RIS model


   - Although the paper claims to tackle weakly supervised referring image segmentation (WSRIS), it raises concerns since both the depth map generator and mask generator are pretrained models that have already been fully supervised to produce segmentation-like maps. This raises the question of whether the problem is genuinely being addressed in a weakly supervised manner.
   - While the mask generator has also been used in baseline algorithms, there is growing concern that weakly supervised learning algorithms are increasingly relying on powerful off-the-shelf models, which could distort the intended fairness of weak supervision.


- Performance Gain from DINOv2
   - The paper additionally employs DINOv2, whereas comparison baselines rely solely on CLIP. It remains unclear whether the performance improvement stems from the proposed method itself or simply from using a more powerful vision encoder.
- Dependency on Off-the-Shelf Pretrained Models
   - Related to the above point, the method’s reliance on a depth map generator—which is essential for computing 3D coordinates—raises fairness concerns when compared with existing methods that do not require depth information. A fair comparison would require experiments without the depth map, although this appears challenging given that the depth map generator serves as a core module in the proposed framework.
   - Similarly, the language consistency module depends on an LLM (Large Language Model), suggesting that solving the WSRIS problem involves leveraging a significantly larger amount of labeled data than typical weakly supervised settings.


- Computational Cost During Training and Inference


   - The approach introduces additional computational and memory costs due to the use of depth map generation, double vision encoder forwarding, and LLM-based processing during both training and inference. A detailed report on these computational overheads is necessary.

**Questions:**

Please provide your responses with reference to the weaknesses mentioned above.

---

### Official Review · Reviewer_qWGf · 2025-10-31

**Soundness:** 2
**Presentation:** 2
**Contribution:** 2
**Rating:** 2
**Confidence:** 5

**Summary:**

This paper tackles the challenging task of Weakly Supervised Referring Image Segmentation (WSRIS), where models must localize and segment target objects referred to in natural language descriptions without pixel-level supervision. The proposed framework, ES3Net, explicitly models semantic-spatial synergy through three components: Explicit Spatial Enhancement Module (ESEM), Language Consistency Module (LCM), Confidence-Aware Dense Distillation (CADD). The method demonstrates strong performance on RefCOCO, RefCOCO+, and G-Ref under both ST and MT supervision settings, outperforming prior SOTA models by a notable margin.

**Strengths:**

- Robustness to linguistic variation: The LCM module addresses the common challenge of paraphrase robustness by enforcing semantic alignment across LLM-generated variants.
- Performance: The method achieves impressive results under both ST and MT supervision, surpassing several SOTA baselines by large margins.
- Clear writing and visuals: The paper is generally well-written with informative diagrams (e.g., Fig. 2–4) that clarify the contributions.

**Weaknesses:**

- Over-Complexity & Over-Reliance on Heuristics. The method combines several moving parts (SAM, Detic, CLIP, DINOv2, pseudo-depth, LLM-generated rephrasings, CADD), which while effective, makes the pipeline heavily dependent on heuristics and black-box models. It raises concerns about generalizability and reproducibility under different image domains or without these specific tools.
- Limited discussion on failure cases or limitations. There’s no clear discussion or analysis of when or why the model might fail (e.g., ambiguous expressions, noisy masks, depth inaccuracies). Qualitative examples are mostly positive.
- Insufficient justification for certain design choices. The use of separate embeddings from CLIP and DINOv2 is shown to work well, but why not use unified multi-modal encoders? Also, why is the centroid depth normalized linearly? Are there alternatives? These architectural decisions could be better motivated.
- MT supervision boundary is unclear. While the paper distinguishes between ST and MT settings, it still uses MT grouping in training under the pretext of no “extra training”. This blurs the weak supervision boundary — it would help to clarify how much this MT prior actually leaks supervision (e.g., in real-world settings, this grouping is rarely available).
- Lack of broader comparisons. The method is not compared against recent LLM+vision approaches (e.g., BLIP-2 or MiniGPT-4 for RIS or grounding). Given LLM usage in LCM, such comparisons would help position the work.

**Questions:**

See weaknesses.

---

### Official Review · Reviewer_LQ8a · 2025-10-31

**Soundness:** 3
**Presentation:** 2
**Contribution:** 2
**Rating:** 4
**Confidence:** 3

**Summary:**

This paper proposes an Explicit Spatial Enhancement Module that explicitly injects geometric context into the embedding space to achieve semantic–spatial synergy. In addition, the authors introduce a Language Consistency Module (LCM) to improve robustness against diverse referring expressions. These components are effectively applied to weakly supervised referring image segmentation (WSRIS), and the proposed approach achieves state-of-the-art performance across multiple RIS benchmarks.

**Strengths:**

1. The ESEM is first approach to incorporate 3D geometric information into WSRIS model.
2. In terms of performance, the method achieves state-of-the-art results across various benchmarks with significant performance gains.

**Weaknesses:**

1. LCM and CADD appear to provide limited novelty and technical contribution, especially when compared to existing techniques used for expression augmentation and knowledge distillation.

2. Moreover, the description of LCM is very brief and only appears at a high level in the overview. In addition, the explanation of CADD lacks sufficient detail, and these components are not clearly illustrated in the figures. As a result, it becomes unclear what the distinct contributions of each module actually are.

3. I understand the idea of injecting geometric information into the representative tokens of each mask. These tokens still need to interact with vision tokens to obtain the similarity score. However, the vision tokens do not contain any geometric information. It remains unclear through what mechanism the spatial cues are propagated during this interaction. More explanation on this part would help clarify the method.

4. The method relies on a mask generator and a depth estimator, which could become an inherent limitation of the approach. However, this paper does not discuss this dependency

**Questions:**

1. In Table 2, does the ablation of ESEM only remove the ‘3D centroid extraction’ component? If this is intended to represent the removal of the full ESEM module, additional results are needed to isolate the effect of removing only the 3D centroid extraction.

2. What is the computational overhead introduced by the mask generator and depth estimator in terms of parameters, FLOPs, and latency?

---

### Note · Authors · 2025-11-14

I have read and agree with the venue's withdrawal policy on behalf of myself and my co-authors.